# Understanding Parinaud’s Syndrome

**DOI:** 10.3390/brainsci11111469

**Published:** 2021-11-06

**Authors:** Juan Fernando Ortiz, Ahmed Eissa-Garces, Samir Ruxmohan, Victor Cuenca, Mandeep Kaur, Stephanie P. Fabara, Mahika Khurana, Jashank Parwani, Maria Paez, Fatima Anwar, Hyder Tamton, Wilson Cueva

**Affiliations:** 1California Institute of Behavioral Neuroscience & Psychology, Faifield, CA 94534, USA; fatimaanwar47@gmail.com; 2School of Medicine, Colegio de Ciencias de la Salud, Universidad San Francisco de Quito, Quito 170901, Ecuador; aeissag@estud.usfq.edu.ec (A.E.-G.); vdcuenca@estud.usfq.edu.ec (V.C.); 3Neurology Department, Larkin Community Hospital, South Miami, FL 33143, USA; ruxmohan@yahoo.com (S.R.); htamton@gmail.com (H.T.); wcueva@gmail.com (W.C.); 4Government Medical College, Patiala 147001, India; mndeep92@yahoo.com; 5School of Medicine, Colegio de Ciencias de la Salud, Universidad Católica de Santiago de Guayaquil, Guayaquil 090615, Ecuador; stephanie.fabara@gmail.com; 6Department of Public Health, University of California, Berkeley, CA 94720, USA; mahika_khurana@berkeley.edu; 7School of Medicine, Neurology Department, Lokmanya Tilak Municipal Medical College, Mumbai 4000022, India; parwanijashank@gmail.com; 8School of Medicine, Colegio de Ciencias de la Salud, Pontificia Universidad Católica del Ecuador, Quito 170143, Ecuador; mochis_9313@hotmail.com

**Keywords:** neurology, parinaud, midbrain, collier sign, pseudo argyll robertson pupil

## Abstract

Parinaud’s syndrome involves dysfunction of the structures of the dorsal midbrain. We investigated the pathophysiology related to the signs and symptoms to better understand the symptoms of Parinaud’s syndrome: diplopia, blurred vision, visual field defects, ptosis, squint, and ataxia, and Parinaud’s main signs of upward gaze paralysis, upper eyelid retraction, convergence retraction nystagmus (CRN), and pseudo-Argyll Robertson pupils. In upward gaze palsy, three structures are disrupted: the rostral interstitial nucleus of the medial longitudinal fasciculus (riMLF), interstitial nucleus of Cajal (iNC), and the posterior commissure. In CRN, there is a continuous discharge of the medial rectus muscle because of the lack of inhibition of supranuclear fibers. In Collier’s sign, the posterior commissure and the iNC are mainly involved. In the vicinity of the iNC, there are two essential groups of cells, the M-group cells and central caudal nuclear (CCN) group cells, which are important for vertical gaze, and eyelid control. Overstimulation of the M group of cells and increased firing rate of the CCN group causing eyelid retraction. External compression of the posterior commissure, and pretectal area causes pseudo-Argyll Robertson pupils. Pseudo-Argyll Robertson pupils constrict to accommodation and have a slight response to light (miosis) as opposed to Argyll Robertson pupils were there is no response to a light stimulus. In Parinaud’s syndrome patients conserve a slight response to light because an additional pathway to a pupillary light response that involves attention to a conscious bright/dark stimulus. Diplopia is mainly due to involvement of the trochlear nerve (IVth cranial nerve. Blurry vision is related to accommodation problems, while the visual field defects are a consequence of chronic papilledema that causes optic neuropathy. Ptosis in Parinaud’s syndrome is caused by damage to the oculomotor nerve, mainly the levator palpebrae portion. We did not find a reasonable explanation for squint. Finally, ataxia is caused by compression of the superior cerebellar peduncle.

## 1. Introduction

In Parinaud’s syndrome or dorsal midbrain syndrome, the two structures that are mainly and causally involved in the symptomatology are the midbrain and the pineal gland. The midbrain can be divided into two segments: the tegmentum or ventral portion and the tectum or dorsal portion. Longitudinally, the midbrain has three main parts: the tectum, tegmentum, and basis. In the tectum is located posteriorly and contains the quadrigeminal plate, gray matter, and fiber tracts [1]. More centrally, we find the tegmentum that contains the cranial nerve nuclei, ventral tegmental area, periaqueductal gray matter, red nucleus, and fiber tracts [1]. The most ventral part, the bases, comprises the substantia nigra, crus cerebri, cerebral peduncles, and corticobulbar fibers [1]. The superior and inferior colliculi are located dorsally. The oculomotor nucleus is located more ventrally than the trochlear nucleus [1]. Longitudinally, it can be divided by the superior and inferior colliculus [2].

The pineal gland weighs 0.1 grams and is part of the epithalamus [3]. The pineal gland plays an important role in the circadian rhythm. It does this by producing melatonin [4]. The main cells of the pineal gland are the pinealocytes, which are parenchymal cells. The pineal gland is surrounded by stromal cells [3]. It is bordered ventrally by the midbrain and quadrigeminal plate, posteriorly by the splenium of the corpus callosum, rostrally by the third ventricle, and caudally by the vermis of the cerebellum [5].

Parinaud’s syndrome is more commonly produced when tumors compress structures around the midbrain, most specifically the tectum of the midbrain. The main manifestations of Parinaud’s syndrome are upward gaze palsy, lid retraction (Collier’s sign), convergence retraction nystagmus (CRN), and pupillary light-near-dissociation [3,5]. However, the pathophysiology related to these signs is mainly unknown and poorly described in the literature.

This study covers the differential diagnosis and the etiologies of Parinaud’s syndrome, follow by a review of the pathophysiology of the signs and symptoms of Parinaud’s syndrome.

## 2. Materials and Methods

For this review, we start with a brief discussion of the differential diagnosis of Parinaud’s syndrome and its main etiologies, and then we proceed to review the pathophysiology of the sign and symptoms of the disease. Table 1 shows the search terms used to gather information for this review.

## 3. Discussion

### 3.1. Differential Diagnosis of Midbrain Syndromes

Syndromes of the midbrain occur mainly due to infarcts or by displacement by tumor masses. Midbrain infarcts occur due to infarcts in the posterior circulation and constitute are 2% of all type of strokes [6]. Benedikt’s, Weber’s, and Claude’s syndrome occur mainly due infarction of arteries of the posterior circulation. In contrast, Nothnagel syndrome is caused by a mass effect of a tumor of the pineal gland. 

Weber’s syndrome is commonly caused by an infarct of the paramedian mesencephalic or the peduncular perforating arteries, branches of the posterior cerebral artery. The main clinical features are oculomotor palsy and contralateral hemiparesis [7]. Contralateral ataxia, 3th nerve palsy, and parkinsonism may be seen if the substantia nigra and cerebellar peduncles are involved [7]. Benedikt’s syndrome is very similar to Weber’s syndrome, except for the development of a tremor. The tremor seen in Benedikt’s syndrome is caused by red nucleus’ involvement, which causes contralateral choreoathetosis and a rubral tremor [8]. In Claude’s syndrome there is no involvement of the corticospinal fibers as opposed to Benedikt’s and Weber’s syndrome. The main symptoms in Claude’s syndrome are contralateral ataxia, and 3th nerve palsy [7].

Unlike the other three syndromes mentioned before, Nothnagel syndrome is caused by a mass effect, analogous to Parinaud [8]. The clinical presentation is very similar to Claude’s syndrome. However, the ataxia is ipsilateral Nothnagel and contralateral in Claude’s [6,9]. Table 2 shows the differential diagnosis of Parinaud’s syndrome.

### 3.2. Etiologies

A pineal mass is the most common cause of Parinaud’s syndrome. Tumors of the pineal gland represent 0.5% of all tumors in adults and 1.9% of those in children [10]. The most common tumor type in the pineal gland is a germ cell tumor, with a frequency of 75% [11]. Besides a pineal mass, other etiologies have been identified in Parinaud’s syndrome. Table 3 describes the various etiologies of Parinaud’s syndrome [12,13,14,15].

### 3.3. Pathophysiology of Parinaud’s Syndrome

#### 3.3.1. Upward Gaze Palsy

Upward gaze palsy is a classical sign of Parinaud’s syndrome. Three structures are mainly involved in vertical gaze: rostral interstitial nucleus of the medial longitudinal fasciculus (riMLF), located laterally to the medial longitudinal fasciculus and rostral to the oculomotor nerve; the interstitial nuclei of Cajal (iNC), located more centrally; and the posterior commissure [16,17]. The riMLF controls vertical smooth pursuit, and torsional saccades, and the iNC serves as a neuronal integrator. The inputs of these nuclei are sent to the final pathway of nucleus III and IV [17]. Activation of the neurons within the riMLF facilitates vertical saccades [16]. 

The riMLF projects ipsilaterally to the inferior rectus subdivision of the oculomotor nerve and bilaterally to the superior rectus subdivisions of the oculomotor nuclei, which generates upward gaze [18,19]. The iNC receive inputs from the contralateral and ipsilateral riMLF and projects their fibers through the posterior commissure [18]. The posterior commissure also receives projections from the frontal eye fields and superior colliculus, and projected to the contralateral riMLF [19].

When there is a dysfunction in the riMLF and iNC, it affects vertical saccades and smooth pursuit [16,19]. Meanwhile interruption of the posterior commissure affects the vestibulo-ocular reflex. The bilateral innervation explains why upward gaze is more affected than downward gaze in Parinaud’s syndrome [19]. Figure 1 shows a diagram of the mechanism of upward gaze palsy.

#### 3.3.2. Convergence Retraction Nystagmus

Convergence retraction nystagmus (CRN) is characterized by deconjugate horizontal jerk nystagmus. CRN is a sign specific to midbrain dysfunction [20]. In the rapid phase of the nystagmus, both eyes converge quickly [20]. Meanwhile, in the slow phase of the nystagmus, both eyes diverge slowly. CRN arises from a dysfunction in the riMLF and the PC [20]. This sign is particularly noticeable in upward gaze [21].

CRN was initially cataloged as a saccadic anomaly. However, Rambolt et al. concluded in their study that it is a vergence anomaly [20]. The nuclei for convergence and divergence are located dorsolateral to the oculomotor nucleus (posterior part of the midbrain) [22]. CRN is caused by damage to the supranuclear fibers, which have an inhibitory effect on the neurons in charge of the convergence and divergence (push-and-pull effect). This lack of inhibition results in a continuous discharge of the medial rectus muscle [22]. Figure 1 shows a diagram of the mechanism of convergence – divergence nystagmus. 

#### 3.3.3. Light-Near Dissociation/Pseudo-Argyll Robertson Pupils

In Parinaud’s syndrome, pupillary constriction in response to light is impaired, but pupillary constriction while convergence and accommodation is preserved [21]. The pretectal nucleus receives afferent projections from the retina and optic nerve via optic tract fibers that bypass the lateral geniculate body [23]. The pretectal nucleus sends a signal to the ipsilateral and contralateral Edinger–Westphal nucleus via the posterior commissure, which contributes to securing the consensual reflex [21,24]. This decussation implies greater susceptibility to the compressive effects of mass lesions, and when there is damage to the cells of the pretectal region, Pseudo-Argyll Robertson pupils are generated [24,25]. In Parinaud’s syndrome the dorsal pretectal area is affected. Anatomically, the ventral area of the pretectal region contains the fibers for accommodation and vergence; on the other hand, the dorsal area contains the light reflex fibers; this fact would explain why pupillary constriction for accommodation is unaffected [21].

When patients have a bilateral lesion of the pretectal nucleus, they show no response to light, and their pupils are considered Argyll Robertson pupils. In patients with Parinaud’s syndrome, there is a small reaction to light when light intensity increases; therefore, it has been called pseudo-Argyll Robertson [25]. According to Binda et al., this phenomenon occurs because an additional pathway for the pupillary response to light that involves conscious attention to a light stimulus, either bright or dark light. This pathway by-passes the pretectal circuit, which explains why these patients retain a slight pupillary response with a strong light stimulus. One hypothesis suggests that melanopsin signals capable of directly activating the Edinger–Westphal nucleus are formed because focused attention enhances the representation of visual stimuli in the occipital cortex [24]. The other possibility is that light exerts down-regulation of sympathetic inhibition to the Edinger–Westphal nucleus, ultimately generating miosis [24]. Figure 2 shows the graphic of Pseudo-Argyll Robertson pupils. 

#### 3.3.4. Collier’s Sign

Collier’s sign refers to upper eyelid retraction that occurs due to lesions in a group of cells near the iNC or the posterior commissure [21]. The posterior commissure is located below the pineal gland. Its main functions are to elicit the pupillary reflex, support vertical eye movement, and inhibit the upper eyelid elevation pathway [21]. Furthermore, in the vicinity of the iNC, there are two groups of cells, the M group and central caudal nuclear (CCN), that elevate the eyelid [21].

The M-group sends excitatory output to the superior rectus (SR), inferior oblique (IO) muscles, and facial nucleus (frontalis muscle), which generate eyelid elevation. Additionally, the M group sends excitatory output to the CCN group of cells which produces eyelid elevation. The CCN produces eyelid elevation by maintaining a constant tonic level of activity. The CCN contributes fibers to the CN III and levator palpebrae superioris allowing elevation of the eyelid. [26].

Lesions in the iNC tend to alter these groups of cells. Two hypotheses have been described for Collier’s sign: The first, overstimulation or under inhibition of the M group [26]. Lesions in the midbrain may trigger overstimulation of the M group, increasing the output to the CCN and allowing more eyelid elevation [26]. The other hypothesis involves an inhibitory nucleus of the posterior commissure (nPC). A lesion in the inhibitory nuclei of the posterior commissure nPC decreases the M group’s inhibition, which increases the excitatory output to the CCN [26].

In an experimental study of macaque monkeys, vertical gaze palsy and upper eyelid retraction occurred when the posterior commissure was damaged. The experiment demonstrated the importance of the posterior commissure on vertical gaze and eyelid control [27]. Vertical gaze palsy can occur with other lesions in the midbrain, especially the tectum. However, the posterior commissure needs to be involved in causing upper eyelid retraction [27]. Figure 3 shows the mechanism of collier sign in patients with Parinaud syndrome. 

#### 3.3.5. III, IV, and VI Nerve Palsies

While the mechanism of III and IV nerve palsies are obvious, the occurrence of VI (abducens) nerve palsy is confusing because the VI nerve is located in the pons [12]. III and IV nerve palsy are related to a mass effect in the midbrain, while VI palsy occurs due to increased intracranial pressure. 

The pineal gland, when enlarges, can compress the Silvian aqueduct, which increases the intracranial pressure causing VI nerve palsy, either unilateral or bilateral [28].

#### 3.3.6. Blurred Vision and Visual Field Defects

Blurred vision is experienced by 25% of patients suffering from Parinaud’s syndrome and it is caused by accommodative spasms that make the patient pseudomyopic, thus causing problems with distant vision [29]. An increase in pineal gland mass can compress the third ventricle, causing multiple visual disturbances because of increased intracranial pressure secondary to obstructive hydrocephalus [30]. Accumulation of extracellular fluids and swelling of optic nerve fibers following raised intracranial pressure leads to papilledema. Persistent papilledema results in optic atrophy and visual field deficits [31].

#### 3.3.7. Ptosis and Squint

Two muscles elevate the eyelid: the superior tarsal muscles and the levator palpebrae superioris [32]. They are innervated by the superior cervical ganglion’s sympathetic fibers and the oculomotor nerve, respectively [32]. The ptosis in Parinaud’s syndrome is caused by compression of the oculomotor nucleus, rather than a problem of the sympathetic fibers. The ptosis presents with strabismus when the oculomotor nerve is involved. Both of these symptoms are usually progressive [33]. A relevant clinical vignette to differentiate Horner’s syndrome ptosis from Parinaud’s syndrome is to note that Parinaud’s ptosis is usually worse [24]. We did not find a reasonable explanation for squinting in this syndrome.

#### 3.3.8. Ataxia

Ataxia could be categorized as a symptom or a sign. The patient might describe feeling unstable and uncoordinated. A physical examination would indicate dysmetria and dysdiadochokinesia due to compression of the superior cerebellar peduncles by an increase in the pineal gland mass. In a report of 40 cases with Parinaud’s syndrome, 7.5% of them presented with ataxia [12].

There is a proximate relation between the cerebellum, the cerebral cortex, and the brainstem, and many fibers tracts course between these structures [1]. The cerebellar output pathways involved in motor coordination pass through the superior cerebellar peduncles and decussate in the midbrain, thus making them susceptible to midbrain lesions [34]. These fibers further connect to the red nucleus and thalamus, so in cases of vascular insults or physical compressions to the midbrain, damage to these and the superior cerebellar peduncles would cause hemiataxia [1]. 

Other midbrain syndromes such as Claude and Nothnangel also present with ataxia. 

Ipsilateral third nerve palsy and contralateral ataxia indicate Claude syndromes, and ipsilateral limb ataxia and unilateral or bilateral third nerve paralysis indicates Nothnagel syndrome [35,36].

#### 3.3.9. Diplopia

Lack of coordination between ocular movements leads to diplopia. It is important to differentiate between unilateral and bilateral diplopia as bilateral involvement usually points towards a central cause [37]. Diplopia is the most common symptom of Parinaud’s syndrome [38]. The fibers responsible for vertical gaze cross the midline in the posterior commissure, thus placing them in the midbrain tectum between the pineal gland and cerebral aqueduct. These fibers cross the midline to connect the contralateral oculomotor (superior rectus) and trochlear nerve nuclei [1]. Tumor or infiltration in the area of the posterior commissure leads to dysfunction of fibers passing through, resulting in diplopia [39]. Vertical diplopia is usually encountered in Parinaud’s syndrome due to trochlear nerve involvement [40]. In a few cases of midbrain lesions, skew deviations are also held responsible for causing vertical diplopia [38].

## 4. Conclusions

The main symptoms of Parinaud’s syndrome are diplopia, blurry vision, visual field defects, ptosis, squint, and ataxia. Diplopia is mainly due to involvement of the IV nerve. Blurry vision is related to accommodation problems, while the visual field defects results from chronic papilledema optic neuropathy. The ptosis in Parinaud’s is caused by damage to the oculomotor nerve. We did not find a good explanation for squint. Finally, ataxia is caused by compression of the superior cerebellar peduncle.

Parinaud’s syndrome’s main signs are upward gaze paralysis, upper eyelid retraction, CRN, and pseudo-Argyll Robertson pupils. Two nuclei are involved in upward gaze palsy: riMLF and the iNC. Collier’s sign occurs when there is damage to the posterior commissure, which usually inhibits eyelid upward gaze. Additionally, overstimulation of M-group cells and an increased firing rate of the CCN generate upward gaze. Convergence retraction nystagmus (CRN) is a vergence anomaly that arises from a dysfunction of riMLF and the posterior commissure. In CRN, there is a continuous discharge of the medial rectus muscle because of the supranuclear fibers’ lack of inhibition. External compression of the posterior commissure causes pseudo-Argyll Robertson pupils. Still, there is some response to light because an additional pathway involves attention to a conscious light stimulus either bright or dark light.

## Figures and Tables

**Figure 1 brainsci-11-01469-f001:**
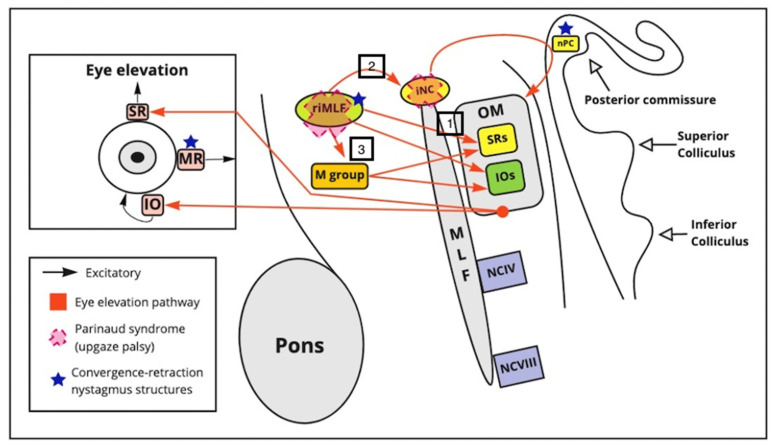
CRN and upward gaze palsy. The superior rectus (SR) and inferior oblique muscle iOM are key for the vertical elevation of the gaze and the oculomotor (OM) nucleus. The riMLF sends information to three destinations: (1) directly to the (SR) and inferior oblique muscle (IOs), (2) to the iNC, which send their axons to the nPC (nucleus of the posterior commissure, (3) to the M group, and CCN which also innervate the RM and iOM nuclei, assisting with upward gaze. CRN is caused by a dysfunction in the riMLF and the nPC, the riMLF receives afferent projections for the center of vergence and convergence. The center of convergence, divergence is located dorsolateral to the oculomotor nerve in the dorsal midbrain.

**Figure 2 brainsci-11-01469-f002:**
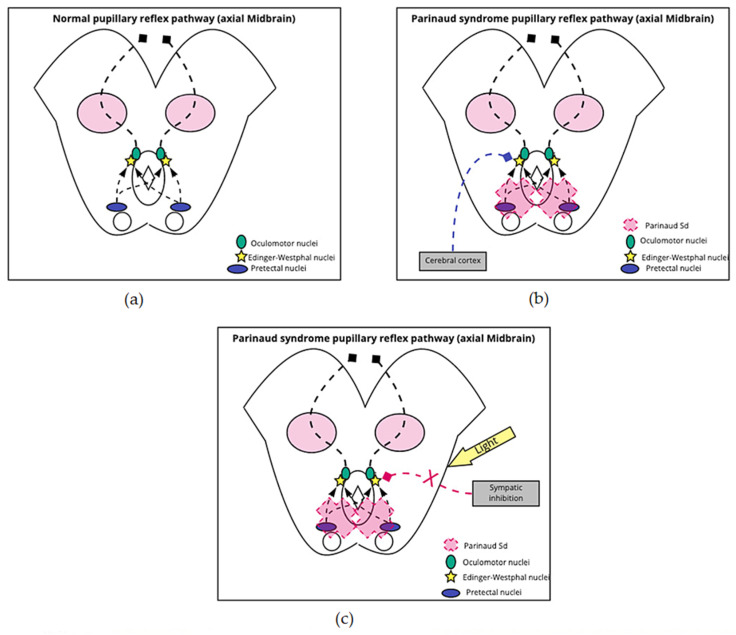
Pseudo-Argyll Robertson Pupils. (**a**) Normal reflex: Signals from the optic nerve enter the pupillary reflex pathway through the pretectal nuclei, which are located anterior to the superior colliculi. The information advances bilaterally to both Edinger–Westphal nuclei, from which exit the parasympathetic fibers that accompany the NC III on its way to the eyeball; (**b**,**c**) Parinaud’s syndrome reflex: The syndrome generates damage to the pretectal nuclei, which prevents signals from reaching the Edinger–Westphal nuclei. To ensure signal arrival to these nuclei and activation of pupillary contraction, the pretectal nuclei can be bypassed and go directly from the cortex; the other possible mechanism is the antagonistic action of light on the sympathetic inhibition of the Edinger–Westphal nuclei.

**Figure 3 brainsci-11-01469-f003:**
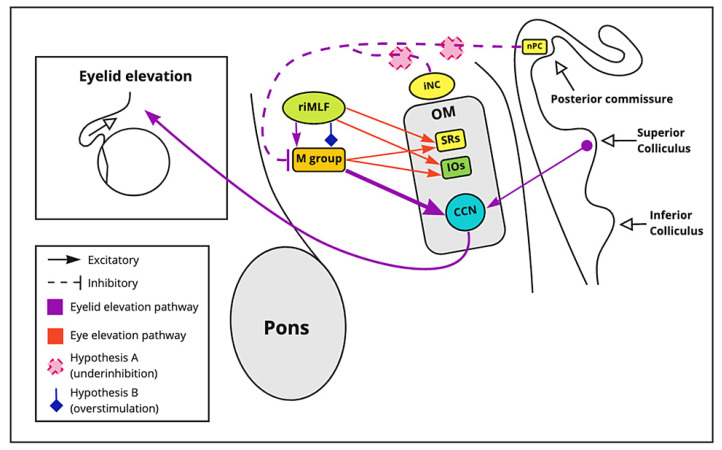
Collier’s sign. The posterior commissure elicits the pupillary reflex, generates the eyes’ vertical movement, and inhibits the upper eyelid elevation pathway. Lesions in the iNC tend to alter M and CCN-group cells. The M-group and riMLF sends excitatory output to the superior rectus (SR), inferior oblique (IO) muscles, and facial nucleus (frontalis muscle), and levator palpebrae which generate eyelid elevation. When M-group cells are stimulated, the firing rate in the CCN increases, as well. Overstimulation or under inhibition of the M group is the proposed hypothesis. Supranuclear control impairment may trigger an overstimulation of the M group by increasing the CCN firing rate. Another hypothesis states that a lesion or damage in the inhibitory nPC and the iNC limits inhibition of the M group. nPC: nuclei of the posterior commissure, iNC: interstitial nucleus of Cajal, riMFL: rostral interstitial nucleus of medial longitudinal fasciculus, OM: oculomotor nuclei, SRs: paramedian nucleus, IOs: intermediate columns nucleus, CCN: central caudal nuclear.

**Table 1 brainsci-11-01469-t001:** Search terms for the information in this review.

Data Search				
(“Parinaud syndrome”[Title/Abstract] OR “dorsal midbrain syndrome”[Title/Abstract]) AND “Diplopia”[Title/Abstract]	(“Parinaud syndrome”[Title/Abstract] OR “dorsal midbrain syndrome”[Title/Abstract]) AND “Ataxia”[Title/Abstract]	(“Parinaud syndrome”[Title/Abstract] OR “dorsal midbrain syndrome”[Title/Abstract]) AND “Nerve Palsy”[Title/Abstract]	“Parinaud syndrome”[Title/Abstract] OR “dorsal midbrain syndrome”[Title/Abstract]) AND “Blurred vision”[Title/Abstract]	(“Parinaud syndrome”[Title/Abstract] OR “dorsal midbrain syndrome”[Title/Abstract]) AND “Ptosis”[Title/Abstract]
(“Parinaud syndrome”[Title/Abstract] OR “dorsal midbrain syndrome”[Title/Abstract]) AND “Collier’s sign”[Title/Abstract]	(“Parinaud syndrome”[Title/Abstract] OR “dorsal midbrain syndrome”[Title/Abstract]) AND “Pseudo Argyll Robertson Pupils”[Title/Abstract]	(“Parinaud syndrome”[Title/Abstract] OR “dorsal midbrain syndrome”[Title/Abstract]) AND “Up gaze Palsy”[Title/Abstract]	(“Parinaud syndrome”[Title/Abstract] OR “dorsal midbrain syndrome”[Title/Abstract]) AND “Converse retraction nystagmus”[Title/Abstract]	(Dorsal midbrain syndrome [Title/Abstract]) AND (Pathophysiology [Title/Abstract])

**Table 2 brainsci-11-01469-t002:** Differential diagnosis of Parinaud’s syndrome.

Midbrain Syndrome	Structures Involved	Clinical Features	Cause
Weber syndrome [1]	Oculomotor nucleus, cerebral peduncle, substantia nigra at times, and superior cerebellar peduncles (ventral tegmentum)	Ipsilateral oculomotor palsy, contralateral ataxia, and contralateral hemiparesis. Parkinsonism is seen at times	Branches of the posterior cerebral artery
Claude syndrome [9]	Red nucleus, oculomotor nucleus, superior cerebellar peduncles	Ipsilateral oculomotor palsy and contralateral ataxia	Branches of the posterior cerebral artery
Benedikt syndrome [8]	Red nucleus, superior cerebellar oculomotor nucleus, cerebral peduncle (paramedial midbrain)	Ipsilateral oculomotor palsy, contralateral cerebellar ataxia, choreoathetosis, and contralateral hemiparesis	Branches of the posterior cerebral artery. Lower frequency of ischemic infarcts compared to Weber and Claude
Nothnagel’s syndrome [6,9]	Superior cerebellar peduncle, oculomotor nucleus (dorsomedial)	Ipsilateral or bilateral oculomotor palsy, ipsilateral limb ataxia	Usually involves a mass compressing the midbrain

**Table 3 brainsci-11-01469-t003:** Etiologies of Parinaud’s syndrome.

Midbrain Infarction	Encephalitis	Trauma	Intracranial Hypotension	Others
Multiple sclerosis	Toxoplasmosis	Obstructive hydrocephalus	Neuromyelitis optica	Miller Fisher syndrome
Midbrain hemorrhage	Cat scratch disease	Tonic clonic seizures	Tectal tuberculoma	Arterio-venous malformations

## Data Availability

Not reported data.

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
