# Peer review of "Understanding Parinaud’s Syndrome"

_brainsci, 2021, doi:10.3390/brainsci11111469_

Round 1

Reviewer 1 Report

Parinaud's syndrome is a cluster of abnormalities of eye movement and pupil dysfunction. It results from injury, either direct or compressive, of the dorsal midbrain causing abnormal motor function of the eye.

This review details the anatomical regions involved in pathology, the potential causes as well as the differential diagnosis and the etiologies. It is extremely well detailed although not always easy to read. The authors analyze the different clinical forms trying to pair with the relative facial nerves.

Author Response

Thank for the wonderful comment, we hired a professional Editor to make to paper easier to read. Additionally, We review some sections of the paper, and improved, to give the reader a better understanding.

1) I explain in the abstract the M group and the central caudal nuclear (CCN) group cells serve as important nuclei for eyelid control and vertical gaze.

2) I clarified in the abstract that points that the ptosis in Parinaud's syndrome occurs mainly.

3) I pointed out the tectum is located posteriorly and made the appropriate grammatical correction. 

4) I added that the tegmentum includes the red nucleus and substantial nigra, which was not included in the original version. 

5) I Changed to subtitle from differential diagnosis to differential diagnosis of midbrain syndromes. 

6) I changed the term dark light stimulus for light stimulus, either bright or dark, to avoid confusion.

7) The iNC nomenclature is now consistent thought all the paper. 

8) I made a much better description of the paragraphs describing the M and CCN cells.

9) We make a good revision and re-wrote some sentences in 3 sections.

  • The differential diagnosis of Parinaud syndrome: In this section, we explained better each of the midbrain syndromes
  • We expanded the section of upward gaze palsy, explaining more extensively the influence the posterior commissure in vertical gaze.
  • We expanded the section of collier sign, improving the explanation of the M group and the CCN

Reviewer 2 Report

Review: Ortiz et al. – Understanding Parinaud’s Syndrome

General: This is a worthwhile review that seems exhaustive in explanations and literature presentation, however, the text is marred by so many nebulous formulations, non-sequiturs and sloppy omissions/mistakes that it takes a lot of good-will to plough through it. I sympathize with the authors who do not seem to be native English speakers. Therefore, I provide a lot of correction suggestions (more than I usually tolerated) instead of rejecting the paper right away. In any case, the paper needs to be massively re-written in order to bring it into a palatable form of the English language.

Major problems: many issues are explained several times, e.g., iNC for interstitial nucleus of Cajal, others are omitted, e.g., M and CCN groups. Several abbreviations are not explained, especially in the figure legends (Rs, iOM, etc).

The authors also should consider introducing and discussion single (SIFs) and multiple innervated fibers (MIFs) of extraocular muscles

Minor problems:

17: investigated

22: interstitial nucleus of Cajal (not: Cajal’s nucleus)

25: explain “M group”

26: explain “CCN group”

28: define Argyll-Robertson pupils

31: involvement of the trochlear nerve (IVth cranial nerve)

33/34: damage to the oculomotor nerve, mainly the levator palpebrae portion

In my opinion, the Abstract should be self-contained and comprehensible by itself without need to consult the main body of the article.

Key words: capitalize the proper names

40: mainly and causally involved in the symptomatology are…

41: delete “While” à Longitudinally,….

43: “The tectum is located the quadrigeminal plate…” - does not make sense!

48 “Longitudinally, it is divided……..colliculi

50: “..red nucleus (nucleus ruber)..

53: “..important role in circadian rhythm by producing…”

55: Replace “limited” with “bordered”

57: What is a “vermix” ? (vermis????)

64/65: “…Parinaud’s syndrome, followed by a ….”

70/71: “..terms, information content and explanations

74: 3.1. Differential diagnosis of fours syndromes

75: Four syndromes of midbrain lesions occur…….and displacement by tumor masses

76: “infarcts constitute 2%....8% of posterior circulation infarcts (6) (ß not clear what this means: explain). Four syndromes need to be discussed in this context.

80: “..present with unresponsive..”

82: Introduce a paragraph break before “Claude’s syndrome”

84: “..Weber’s syndrome, but has tremor as an added symptom.

85: delete “helps differentiate between these two entities”

86: “is due to red nucleus involvement,..”

88: “….is very similar to Parinaud’s syndrome.”

89: “mainly due to mass displacement effects. However, the symptomatology is different: Nothnagel’s….” 

96: replace “implicated” with “identified”

97: replace “describes” with “lists”

97: delete “Parinaud’s syndrome”

102: “..vertical gaze: rostral interstitial nucleus of the medial longitudinal fasciculus (riMLF)

103 (iNC)

105 iNC

106: “are transmitted to the final output pathways of cranial nuclei III and IV (oculomotor and trochlear nuclei)”

107: what is a “burst” of neurons?

107: iNC

108: “and also projects via the posterior commissure”

109-111 ”The iNC sends……to the inferior rectus subdivision of the oculomotor nucleus, and bilaterally to the superior rectus subdivisions of the oculomotor nuclei.”

111: delete “the”

112: delete “the” (before “upward”)

118: “…from a dysfunction of the riMLF and the PC”

133: explain Argyll-Robertson’s pupils in the beginning of the paragraph

135: replace “with” with “while”

143: “near reflex” ????

148: delete “the”

151: what is a “bright dark” stimulus?

167: delete “the”

168: delete” the”

169: “..Collier’s sign: the posterior commissure and the iNC.”

171: “..pupillary reflex, support vertical eye movements, and…”

173: delete “cells”

173/174: explain M and CCN groups (what are they? Function etc. ? Is the M group meant as a subdivision of the medial rectus motoneuron pool?)

174: “..tend to alter the activity in these..”

175: “..synapses at the oculomotor nerve ?????

176 ff: While M group cells… incomprehensible sentence! – needs re-working!

179 “first”

182: “..damage to the..”

183: “…and iNC limits..”

186-188: incomprehensible sentences

201: “..are obvious..”

201: “the occurrence of VI (abducens) nerve palsy…”

202: “ ..is located in the posterior brainstem”.

204: “due to an increase in pineal gland mass..”

205: “..the abducens nerve is most commonly affected.

209: ”..and it is caused..”

210: “An increase in pineal gland mass can compress…

213: “..papilledema. Persistent papilledema….deficits such as enlargement of..”

229: “..due to compression of the superior cerebellar peduncles by an increase pineal gland mass. In a report….presented….”

233: replace “relay” with “course”

233: “The cerebellar output pathways involved in motor coordination pass through the superior cerebellar peduncles and decussate in the midbrain, thus...”

235: “Spinothalamic- rubric (rubro) thalamic”???? These are not cerebellar output pathways!! So: delete.

240: replace “include” with “between”

240: “syndromes”

241: delete “,” (comma)

249: “..and the cerebral..”

250/251: “..connect the contralateral oculomotor (superior rectus) and trochlear nuclei. Tumor masses or infiltrations …lead …. of passing fibers,..”

253: “…due to trochlear nerve involvement.

261: “damage to the …”

264: “..upward..”

265: “..nucleus of the…”

267: “…the interstitial nucleus of Cajal (iNC)

274: what is a “conscious bright dark stimulus”?

Figure legends: need critical rewriting, explanations and a list of abbreviations in each.

Author Response

1)I  explained in the abstract the M group and the central caudal nuclear (CCN) group cells serve as important nuclei for eyelid control and vertical gaze.

2) I clarified in the abstract that points that the ptosis in Parinaud's syndrome occurs mainly.

3) I pointed out the tectum is located posteriorly and made the appropriate grammatical correction. 

4) I added that the tegmentum includes the red nucleus and substantial nigra, which was not included in the original version. 

5) I Changed to subtitle from differential diagnosis to differential diagnosis of midbrain syndromes. 

6) I changed the term dark light stimulus for light stimulus, either bright or dark, to avoid confusion.

7) The iNC nomenclature is now consistent thought all the paper. 

8) I produce a much better description of the paragraphs describing the M and CCN cells.

9) We make a good revision and re-wrote some sentences in 3 sections.

  • The differential diagnosis of Parinaud syndrome: In this section, we explained better each of the midbrain syndromes
  • We expanded the section of upward gaze palsy, explaining more extensively the influence the posterior commissure in vertical gaze.
  • We expanded the section of collier sign, improving the explanation of the M group and the CCN